# ScholarCopilot: Training Large Language Models for Academic Writing with Accurate Citations

**Yubo Wang**[1,4,†], **Xueguang Ma**[1,†], **Ping Nie**[3], **Huaye Zeng**[1], **Zhiheng Lyu**[1], **Yuxuan Zhang**[1], **Benjamin Schneider**[1], **Yi Lu**[1], **Xiang Yue**[2], **Wenhu Chen**[1,4,†]
[1]University of Waterloo, [2]Carnegie Mellon University, Pittsburgh,
[3]Independent Researcher, [4]Vector Institute, Toronto

## Abstract

Academic writing requires both coherent text generation and precise citation of relevant literature. Although recent Retrieval-Augmented Generation (RAG) systems have significantly improved factual accuracy in general-purpose text generation, their ability to support professional academic writing remains limited. In this work, we introduce ScholarCopilot, a unified framework designed to enhance existing large language models for generating professional academic articles with accurate and contextually relevant citations. ScholarCopilot dynamically determines when to retrieve scholarly references by generating a retrieval token [RET], which is then used to query a citation database. The retrieved references are fed into the model to augment the generation process. We jointly optimize both the generation and citation tasks within a single framework to improve efficiency. Our model is built upon Qwen-2.5-7B and trained on 500K papers from arXiv. It achieves a top-1 retrieval accuracy of 40.1% on our evaluation dataset, outperforming baselines such as E5-Mistral-7B-Instruct (15.0%) and BM25 (9.8%). On a dataset of 1,000 academic writing samples, ScholarCopilot scores 16.2/25 in generation quality—measured across relevance, coherence, academic rigor, completeness, and innovation—significantly surpassing all existing models, including much larger ones like the Retrieval-Augmented Qwen2.5-72B-Instruct. Human studies further demonstrate that ScholarCopilot, despite being a 7B model, significantly outperforms ChatGPT, achieving 100% preference in citation quality and over 70% in overall usefulness.

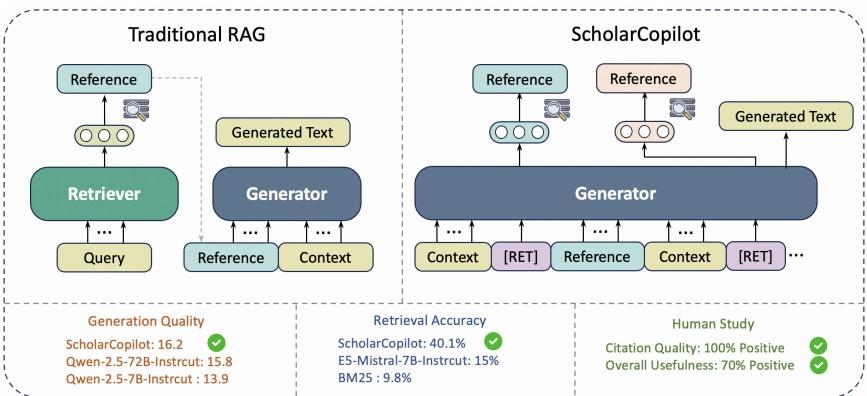

Figure 1: Comparison of traditional Retrieval-Augmented Generation (RAG) systems and our proposed **ScholarCopilot**. Traditional RAG systems (left) separately perform retrieval and generation, leading to representation misalignment. In contrast, ScholarCopilot (right) dynamically generates retrieval tokens ([RET]) during text generation for integrated and context-aware reference retrieval.

---

[0]† Core Contributors
[1]† Project website: https://tiger-ai-lab.github.io/ScholarCopilot/

# 1 Introduction

Academic writing is a knowledge-intensive task that requires both structured content generation and accurate citation of relevant literature. While large language models (LLMs) such as GPT-4 (Achiam et al., 2023), Deepseek-v3 (Liu et al., 2024), and Qwen2.5 (Yang et al., 2024) can generate fluent academic-style text, they frequently hallucinate citations, undermining their reliability for research writing (Huang et al., 2025; Tonmoy et al., 2024).

Recent advanced retrieval-augmented generation (RAG) (Lewis et al., 2020; Shi et al., 2023) systems address this issue by retrieving relevant references from external knowledge bases to enhance factual consistency and reduce hallucinations. These approaches typically follow a **first-retrieve-then-generate** pipeline, as illustrated in Figure 2 (left), where retrieval is conducted independently prior to generation. However, this pipeline neglects the evolving generation context, making it difficult to dynamically adjust retrieval decisions based on the changing information needs during writing. For instance, when generating an introduction mentioning GPT-4, traditional approaches cannot adaptively retrieve GPT-4-related references precisely when needed, since retrieval decisions are predetermined without awareness of the specific generation context. Consequently, these methods suffer from three key limitations: (1) separate optimization of retrieval and generation models leads to misalignment in query intent; (2) predetermined retrieval decisions lack flexibility and context-awareness; (3) static pipeline limits user control over the generation of content and citation needs.

To overcome these limitations, we propose **ScholarCopilot**, an agentic RAG framework tailored for assisting academic paper writing that seamlessly integrates text generation and citation retrieval in a unified, iterative manner, as illustrated in Figure 2 (right). Instead of relying on separate retrieval and generation stages, ScholarCopilot dynamically determines when retrieval is necessary by generating special **retrieval tokens ([RET])** based on the evolving generation context. Upon generating these tokens, ScholarCopilot pauses the generation process, retrieves relevant scholarly references, and integrates their content (abstracts or key excepts) directly back into subsequent generation steps. The dense representations of these retrieval tokens are optimized via contrastive learning, enabling efficient similarity search. Additionally, ScholarCopilot allows optional user refinement and citation triggering during the iterative process, providing flexibility to integrate human domain expertise for further improving generation quality. This unified, iterative approach enhances citation accuracy, improves content coherence, and maintains efficiency without additional overhead.

We evaluate ScholarCopilot extensively on academic writing tasks, focusing on **generation quality, retrieval accuracy, and overall user experience**. Our model achieves **40.1% top-1 retrieval accuracy**, significantly surpassing baselines such as **E5-Mistral-7B-Instruct (15.0%)** (Wang et al., 2023) and **BM25 (9.8%)** (Robertson et al., 2009), with consistent performance gains across all Top-K thresholds. In terms of generation quality, ScholarCopilot scores **16.2/25** on a 1000 samples dataset with LLM-as-judge across five dimensions (relevance, coherence, academic rigor, completeness, and innovation), substantially outperforming larger models such as **Qwen-2.5-7B-Instruct (13.9)** and **Qwen-2.5-72B-Instruct (15.8)**. A comprehensive user study with 10 experienced academic writers further confirms ScholarCopilot's effectiveness, particularly highlighting its citation accuracy (100% positive ratings) and overall usefulness (70% positive ratings) compared to ChatGPT.

Our main contributions are summarized as follows:

- **A unified generation-retrieval model** that effectively integrates retrieval into the generative process, enabling seamless citation retrieval while reducing inference overhead and improving citation accuracy and relevance.
- **A comprehensive evaluation framework** that assesses both retrieval accuracy and academic text quality along five critical dimensions: content relevance, logical coherence, academic rigor, information completeness, and scholarly innovation.
- **A large-scale training dataset** consisting of 500k computer science papers from arXiv with comprehensive citation networks (33 matched citations per paper on average), facilitating robust learning of academic writing patterns and citation-aware scholarly practices.

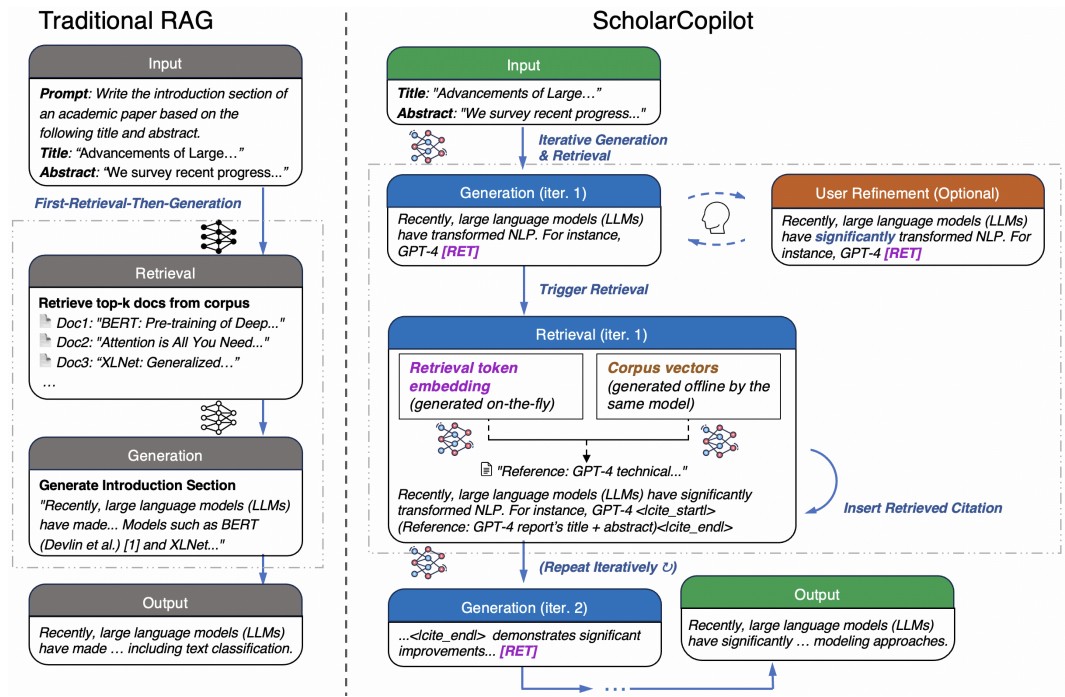

Figure 2: Comparison between traditional Retrieval-Augmented Generation (RAG) methods (left) and ScholarCopilot (right). Traditional RAG follows a static retrieval-then-generation pipeline, retrieving references independently before generation. ScholarCopilot dynamically interleaves retrieval and generation by producing retrieval tokens ([RET]) based on current context, enabling context-aware citation retrieval and optional user refinement.

## 2 Related Work

### 2.1 Dense Retrieval

Recent studies have demonstrated that dense retrieval methods using pretrained language models to encode text into dense vectors outperform traditional lexical retrievers like TF-IDF and BM25. Following the introduction of DPR (Karpukhin et al., 2020), several approaches have been proposed to improve dense retrieval through advanced training strategies (e.g., ANCE (Xiong et al., 2021), Condenser (Gao & Callan, 2021)), data augmentation techniques (e.g., BGE (Xiao et al., 2023), GTE (Li et al., 2023), DRAMA (Ma et al., 2025)), and by leveraging large language models as backbones (e.g., LLM2Vec (BehnamGhader et al., 2024), RepLlama (Ma et al., 2024), Mistral-E5 (Wang et al., 2023)). Today, commercial embedding models (e.g., OpenAI (Neelakantan et al., 2022), GeminiEmbed (Lee et al., 2025)) are widely used in real-world retrieval systems. However, most existing methods are designed for single-turn retrieval with short queries for retrieval, which is not well suited for citation suggestions where paper contexts are used to retrieve the next relevant citation.

### 2.2 Retrieval Augmented Generation

In the era of LLM, the Retrieval Augmented Generation (RAG) paradigm integrates a retrieval model for the generation model, allowing the generation model to have access to external knowledge, improving the generation's correctness and factuality for downstream tasks, such as question answering or fact verification (Petroni et al., 2021). Traditional RAG methods typically follow a retrieve-then-generate pipeline (Lewis et al., 2020; Gao et al., 2024), where retrieval is conducted independently based on an initial query, and the retrieved documents are concatenated as context for the generation model. While effective for short-form generation tasks, this static pipeline struggles in scenarios requiring long-

Figure 3: The pipeline for creating the ScholarCopilot dataset. Our final dataset includes 10M citations matched from arXiv and 6.8M citations matched from Semantic Scholar (one paper may be cited by multiple articles). However, at inference time, to ensure reference quality, we only use the 670K articles from arXiv as the corpus.

form generation with evolving information needs. To address this limitation, recent methods such as FLARE (Jiang et al., 2023) and SelfRAG (Asai et al., 2023) propose iterative RAG strategies, where retrieval and generation are interleaved, allowing retrieval decisions to adapt dynamically based on the generation trajectory. These systems demonstrate improved factual accuracy for long-form content by leveraging the generation context to refine retrieval queries. Recent work, OpenScholar (Asai et al., 2024) aims to improve long-form scientific question answering with self-feedback inference in RAG. However, they still decouple the retrieval and generation models, which can lead to representational misalignment for implicit query intent and increased inference overhead. More unified approaches, such as GritLM (Muennighoff et al., 2025) and OneGen (Zhang et al., 2024), train a unified model to serve both as the generator and retriever. These models share representations and can cache hidden states during generation, improving the efficiency of the system. Despite their advantages, most of these systems have been evaluated primarily on QA-style benchmarks (Mallen et al., 2023) and do not consider the iterative and citation-centric requirements of academic writing.

Our approach differs from prior work in three ways. First, it uses iterative RAG, interleaving retrieval with generation to fit evolving citation needs. Second, it handles implicit intent, inferring citations from context without explicit queries. Third, it enables human-in-the-loop interaction, allowing users to guide or refine citations during writing.

## 3 ScholarCopilot

### 3.1 Dataset

To train a model capable of accurately generating academic text with appropriate citations, we constructed a large-scale dataset of computer science research papers. Our dataset creation process consisted of five major stages, as illustrated in Figure 3.

**Stage 1: Paper Collection.** We collected 670K computer science papers published on arXiv (Ginsparg, 2011) between 2007 and 2024. From this initial corpus, we successfully obtained LaTeX source code for 570K papers, which formed the foundation of our dataset.

**Stage 2: Structure Parsing.** We developed heuristic methods to parse the LaTeX source files and extract structured components, including titles, abstracts, introductions, related work sections, and bibliographies. This stage involved handling complex LaTeX formatting and nested environments. After filtering out papers with parsing failures, we retained 501K successfully structured documents (500K for training and 1K for evaluation), preserving the hierarchical organization essential for understanding academic documents.

**Stage 3: Citation Extraction.** We extracted citation information from bibliographic entries in each paper. Due to the diversity of BibTeX formatting conventions, regular expression-based approaches proved ineffective for reliable title extraction. Instead, we employed the Qwen-2.5-3B-Instruct (Yang et al., 2024) model to robustly extract paper titles from bibliography entries. This approach yielded 19M unique citation titles across our corpus.

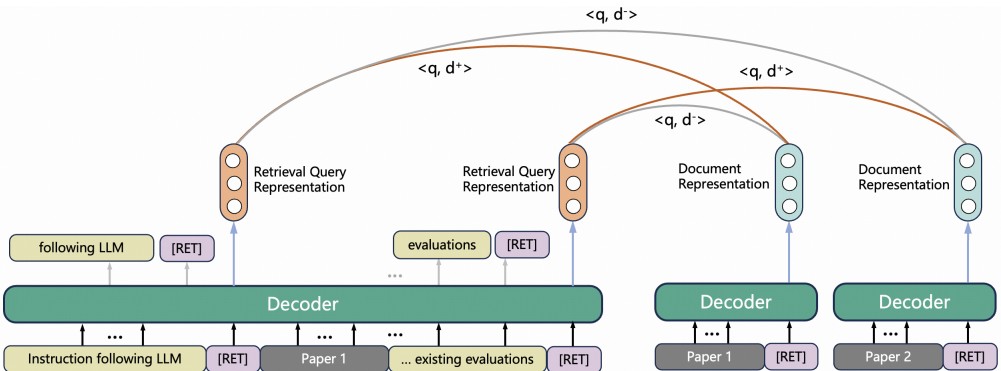

Figure 4: **Unified training framework of ScholarCopilot.** The architecture jointly optimizes the next token prediction loss for text generation and the contrastive loss for citation retrieval. Retrieval tokens ([RET]) dynamically trigger retrieval. $< q, d^+ >$ indicates the positive pair of query and document during contrastive learning, and $< q, d^- >$ indicates the negative pair. The generation model and retrieval model share parameters. In this figure, Paper 1 and Paper 2 can be considered as hard negatives for each other.

**Stage 4: Reference Matching.** To enable retrieval during training and inference, we matched the extracted citation titles against established academic databases. Of the 19M citation titles, we successfully matched 10M in the arXiv metadata repository and an additional 6.8M in the Semantic Scholar database (Kinney et al., 2023), resulting in a total of 16.8M matched citations. The remaining unmatched citations typically corresponded to URLs or publications not indexed in either database.

**Stage 5: Dataset Integration.** Finally, we integrated the parsed paper structures with their matched citations to create the comprehensive ScholarCopilot dataset. The training dataset comprises 500K papers, with 1K papers reserved for evaluation. Each paper contains an average of 38 citations, of which we successfully matched 33 (87%) to their corresponding entries in academic databases.

### 3.2 Unified Training for Generation and Citation Retrieval

ScholarCopilot jointly optimizes two objectives: next token prediction for text generation and contrastive learning for citation retrieval. Figure 4 illustrates this architecture. Detailed training procedures and hyperparameters can be found in the Appendix A.1.

**Next Token Prediction Loss $L_g$.** ScholarCopilot adopts the standard autoregressive language modeling objective for text generation, maximizing the log-likelihood of each token $x_t$ conditioned on previous tokens $x_{<t}$ and retrieved content $c$ (e.g., paper abstracts) when retrieval occurs: $L_g = - \sum_t \log p(x_t | x_{<t}, c)$. Retrieval is dynamically triggered via special tokens ([RET]) generated during inference.

**Contrastive Loss $L_r$ for Citation Retrieval.** To optimize retrieval token representations, ScholarCopilot employs contrastive learning, encouraging higher similarity between retrieval token embeddings $q$ and positive (relevant) citation embeddings $d^+$, and lower similarity with negative (irrelevant) citations $d^-$. Formally, the contrastive loss is defined as $L_r = - \log[\exp(\text{sim}(q, d^+)) / (\exp(\text{sim}(q, d^+)) + \sum_{d^-} \exp(\text{sim}(q, d^-)))]$, where $\text{sim}(\cdot, \cdot)$ denotes cosine similarity. Positive citations are those referenced in the ground-truth paper. Negative citations are obtained through in-batch sampling: citations from the same paper irrelevant to the current context serve as hard negatives, while those from other papers in the batch are easy negatives.

**Joint Optimization.** ScholarCopilot minimizes the combined loss $L_{total} = L_g + \lambda L_r$, where $\lambda$ balances generation and retrieval objectives. In our experiments, we set $\lambda = 1$, equally

weighting both terms. Joint optimization ensures effective retrieval token learning for accurate citation retrieval without compromising generation quality.

## 4   Experiments

### 4.1   Baselines

We compare ScholarCopilot against several baseline approaches. For generation baselines, we include: **Qwen-2.5-7B-re**, Qwen-2.5-7B-Instruct enhanced by citation retrieval using E5-Mistral-7B-Instruct; **Qwen-2.5-72B-re**, the larger 72B parameter variant with the same retrieval method; **Qwen-2.5-7B-gt**, Qwen-2.5-7B-Instruct provided with ground truth citations as input; and **Qwen-2.5-72B-gt**, the 72B parameter variant with ground truth citations. Retrieval baselines include **BM25** (Robertson et al., 2009), a classical lexical retrieval approach commonly used in information retrieval systems; and **E5-Mistral-7B-Instruct** (Wang et al., 2023), a recent embedding-based retrieval model fine-tuned for retrieval tasks.

**Training Details.** ScholarCopilot was trained on 50K training samples for 2 epochs. We used the following hyperparameters: maximum context length of 16,384 tokens, learning rate of $1 \times 10^{-5}$, per-device training batch size of 1, and gradient accumulation steps of 4. Training was performed on 4 machines, each equipped with 8 NVIDIA H100 GPUs, resulting in a global batch size of 128.

### 4.2   Evaluation Methodology

While several existing benchmarks evaluate citation-related tasks, they are not suitable for our academic writing scenario. ALCE (Gao et al., 2023) evaluates LLMs' ability to generate grounded text with citations, but focuses on general question-answering tasks rather than academic writing contexts. CiteMe (Press et al., 2024) addresses citation attribution—determining which cited source supports a given claim—rather than the citation generation task we target. LitSearch (Ajith et al., 2024) constructs a literature search benchmark simulating real research scenarios, but operates in a question-answering format requiring explicit query generation, which differs from our academic writing evaluation where citations must be generated based on contextual flow rather than explicit queries. Therefore, we design our own evaluation tailored for academic writing scenarios.

We evaluate models across two primary criteria: generation quality and retrieval accuracy.

**Generation Quality.** We evaluate academic writing quality using five comprehensive dimensions, each scored from 1 (poor) to 5 (excellent):

*Content Relevance* measures how well the generated text aligns with the specific academic topic, including appropriate use of domain-specific terminology, accurate representation of research concepts, and contextual appropriateness within the broader scientific discourse. *Logical Coherence* assesses the clarity and logical flow of arguments, including proper sequencing of ideas, effective transitional phrases, consistency in reasoning, and overall readability. *Academic Rigor* evaluates scholarly depth and precision, encompassing technical accuracy, appropriate level of detail, proper qualification of claims, and adherence to academic writing conventions. *Information Completeness* measures coverage comprehensiveness, including whether key aspects are addressed, sufficient detail is provided, and no critical information gaps exist. *Scholarly Innovation* assesses originality and insightfulness, including novel perspectives, creative synthesis of existing knowledge, identification of research gaps, and potential contribution to the field.

For evaluation, we provide GPT-4o with the paper's title and abstract as contextual input, along with the ground truth continuation (introduction and related work sections) from published papers and the corresponding model-generated text. The evaluator compares the generated content against the ground truth across all five dimensions, providing both numerical scores and detailed justifications. To ensure reliability, we conducted a validation study where human experts evaluated a subset of 100 examples using the same criteria. The inter-annotator agreement between GPT-4o and human evaluators achieved a Pearson

| Model | Relevance | Coherence | Academic | Completeness | Innovation | Total |
|---|---|---|---|---|---|---|
| | | | Groundtruth Citations | | | |
| Qwen-2.5-7B-gt | 3.27 | 3.07 | 2.52 | 2.77 | 2.82 | 14.44 |
| Qwen-2.5-72B-gt | 3.73 | 3.71 | 3.00 | 3.11 | 3.28 | 16.82 |
| | | | Retrieved Citations | | | |
| Qwen-2.5-7B-re | 3.16 | 3.30 | 2.26 | 2.41 | 2.80 | 13.94 |
| Qwen-2.5-72B-re | 3.56 | 3.61 | 2.68 | 2.84 | 3.12 | 15.81 |
| ScholarCopilot | 3.63 | 3.66 | 2.87 | 2.89 | 3.17 | 16.21 |
| Δ (Ours - 72B) | +0.07 | +0.05 | +0.09 | + 0.04 | +0.05 | +0.4 |

Table 1: Generation quality evaluation results by GPT-4o. All scores are on a scale of 1-5, except for Total which is the sum (max 25).

correlation of 0.78 across all dimensions, indicating strong agreement and falling within the range typically considered highly reliable for subjective assessment tasks in academic evaluation studies. Detailed evaluation prompts are provided in the appendix A.2.

**Retrieval Accuracy.** Citation retrieval is evaluated using Recall@k (k = 1 to 10), defined as the proportion of cases where the correct citation appears among the top-k retrieved results. Specifically, citations and subsequent content in 1,000 test samples are masked, and retrieval models predict citations based solely on the preceding context. For baseline models, we found that using the entire preceding context reduces performance; thus, we only use the last sentence before the citation as the query. Recall@k is computed by comparing predicted citations to the original ground-truth citations.

## 4.3 Main Results

Table 1 presents the generation quality results for our approach compared to baseline models. ScholarCopilot achieves a total score of 16.21 out of 25, outperforming both Qwen-2.5-7B-Instruct with retrieval enhancement (13.94) and standard Qwen-2.5-7B-Instruct with ground truth citations (14.44). Notably, our approach even surpasses the much larger Qwen-2.5-72B-Instruct model with retrieval enhancement (15.81) and comes close to the 72B model with ground truth citations (16.82), despite having only about 10% of its parameters.

ScholarCopilot demonstrates particular strengths in Relevance (3.63) and Coherence (3.66), comparable to the 72B models. The improvement in Academic Rigor (2.87 vs. 2.26 for Qwen-2.5-7B-re) highlights our model's ability to incorporate appropriate citations and scholarly conventions. These results confirm that our unified approach to generation and citation effectively improves academic writing quality even with a relatively small model.

## 4.4 Ablation Studies

### 4.4.1 Retrieval Performance

Figure 5 compares ScholarCopilot's citation retrieval performance (Recall@k) with baseline methods, including OpenScholar's retriever both with and without a reranker. ScholarCopilot achieves a top-1 recall of 40.1%, significantly outperforming BM25 (9.8%) and E5-Mistral-7B-Instruct (15.0%). While OpenScholar's retriever achieves competitive performance, particularly when enhanced with a reranker (24.6% at R@1), ScholarCopilot still maintains a substantial advantage. This advantage persists across all recall levels, with ScholarCopilot reaching 64.8% recall@10, outperforming OpenScholar + Reranker (48.6%), more than doubling E5-Mistral-7B-Instruct (30.0%), and tripling BM25 (20.8%).

These results demonstrate that while OpenScholar performs reasonably well in our evaluation setting, especially when combined with a reranker, our approach maintains superior performance. Importantly, OpenScholar is primarily designed for QA tasks and trained on question-answering data, whereas our task focuses on autocomplete for scientific writing.

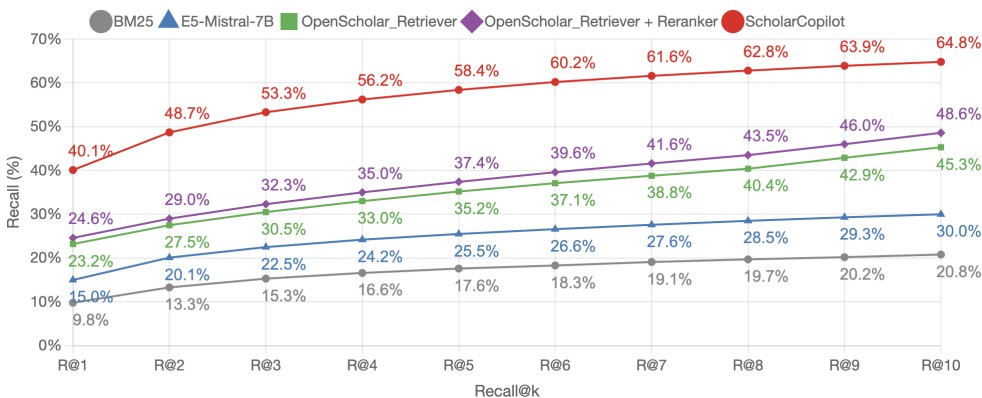

Figure 5: Comparison of citation retrieval performance (Recall@k) between ScholarCopilot and baseline retrieval methods (BM25, E5-Mistral-7B-Instruct and OpenScholar).

The consistent advantage of ScholarCopilot across all recall levels highlights the effectiveness of our unified training approach. Traditional retrieval methods rely on explicitly formulated queries, often failing to capture nuanced citation intents. In contrast, ScholarCopilot directly optimizes retrieval token representations during generation, implicitly encoding citation intent through context-aware queries informed by both local (surrounding text) and global (document-level) information.

### 4.4.2 Impact of Reference Content Integration

We evaluate the impact of providing retrieved reference content to ScholarCopilot during generation by comparing two settings: (1) the standard approach, where the model accesses reference details during generation; and (2) a variant that triggers retrieval but cites papers without seeing their content.

| Method | Relevance | Coherence | Academic | Completeness | Innovation | Total |
|---|---|---|---|---|---|---|
| ScholarCopilot | 3.63 | 3.66 | 2.87 | 2.89 | 3.17 | 16.21 |
| w/o ref. content | 3.60 | 3.25 | 2.58 | 2.91 | 3.19 | 15.53 |
| Δ | +0.03 | +0.41 | +0.29 | -0.02 | -0.02 | +0.68 |

Table 2: Impact of reference content integration on generation quality.

As shown in Table 2, the two variants perform similarly on Relevance, Completeness, and Innovation. However, differences appear in Coherence (3.66 vs. 3.25) and Academic Rigor (2.87 vs. 2.58), leading to a higher total score for the standard ScholarCopilot (16.21 vs. 15.53). Analysis indicates two reasons. First, access to reference content reduces inaccuracies when describing cited works. Second, reference details provide contextual information that improves coherence, especially during comparisons or transitions between ideas.

Qualitative evaluation shows the variant without reference content tends to cite sources with general statements, whereas the standard approach integrates specific details for clearer connections. For example, when discussing neural networks, the standard model states: "Transformer models leverage self-attention mechanisms to capture long-range dependencies `cite(vaswani2017attention)`, specifically through a multi-headed approach that projects queries, keys, and values into separate subspaces." In contrast, the variant without reference content produces simpler statements like "Transformer models use self-attention for capturing dependencies `cite(vaswani2017attention)`."

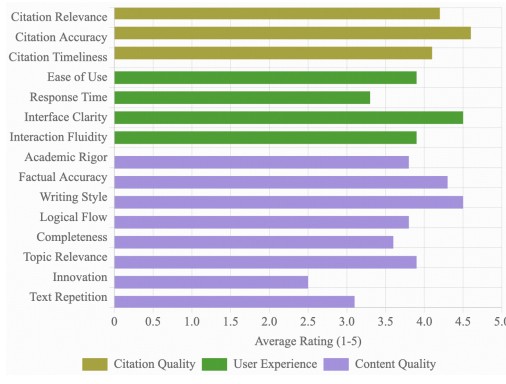
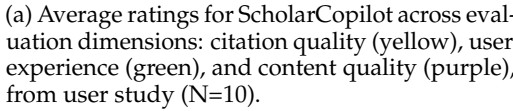

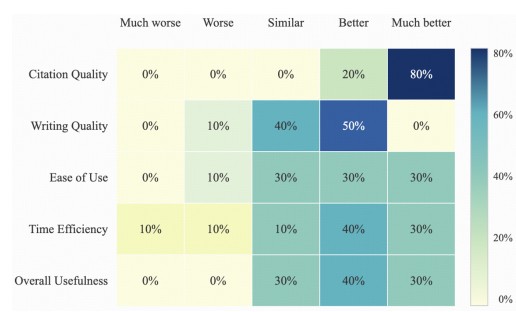

(a) Average ratings for ScholarCopilot across evaluation dimensions: citation quality (yellow), user experience (green), and content quality (purple), from user study (N=10).

(b) Comparative analysis of ScholarCopilot vs. ChatGPT across five dimensions: Citation Quality, Writing Quality, Ease of Use, Time Efficiency, and Overall Usefulness. Darker blue indicates higher percentages of ratings.

Figure 6: Human evaluation of ScholarCopilot and comparative analysis with ChatGPT

## 5 User Study

To evaluate the utility of ScholarCopilot in practical academic writing, we conducted a user study with participants from various academic backgrounds. This evaluation assessed both technical performance and user experience.

### 5.1 Human Evaluation Design

We conducted a mixed-method evaluation combining quantitative ratings and qualitative feedback. Participants were 10 students (5 PhD, 4 master's, and 1 undergraduate), averaging 4.2 years of academic writing experience. All participants had prior academic writing experience and were familiar with AI writing assistants such as ChatGPT.

Each participant used ScholarCopilot to draft the introduction and related work sections on at least five topics within their expertise. The evaluation included:
**Quantitative Assessment.** Participants rated ScholarCopilot on 15 metrics using a 5-point Likert scale (1=Poor, 5=Excellent), grouped into Citation Quality (relevance, accuracy, timeliness), User Experience (ease of use, response time, interface clarity, interaction fluidity), and Content Quality (academic rigor, factual accuracy, writing style, logical flow, completeness, topical relevance, innovation, redundancy).
**Comparative Analysis.** Participants compared ScholarCopilot with ChatGPT on citation quality, writing quality, ease of use, time efficiency, and overall usefulness.
**Open-ended Feedback.** Participants commented on ScholarCopilot's strengths, limitations, and suggested improvements.

### 5.2 Human Evaluation Results

Figure 6a shows the average ratings. ScholarCopilot received the highest scores for citation accuracy (4.6/5), interface clarity (4.5/5), and writing style (4.5/5). Citation quality metrics averaged 4.3/5. User experience metrics averaged 3.9/5, with response time rated lowest (3.3/5). It is worth noting that the system was deployed on a single 80GB GPU, which led to longer waiting times during peak usage periods. Due to this resource limitation, different participants experienced significantly varied response times, which explains the inconsistent feedback regarding system responsiveness in the evaluation results.

Content quality metrics showed more variation, with Writing style (4.5/5) and factual accuracy (4.3/5) scoring well, while innovation received the lowest score across all metrics (2.5/5). This suggests that while ScholarCopilot excels at generating academically sound content, it may be less effective at proposing novel ideas or suggesting innovative directions.

**Comparative Advantage.** Figure 6b compares ScholarCopilot and ChatGPT. ScholarCopilot shows a clear advantage in citation quality, with all participants rating ScholarCopilot higher. For overall usefulness, 70% rated ScholarCopilot higher. Writing quality advantage was moderate, with 50% rating ScholarCopilot higher and 40% assessing it similar to ChatGPT.

**Qualitative Feedback.** Open-ended responses identified strengths such as integrated citation management (citation search and BibTeX handling), interactive incremental writing style offering greater user control, and improved time efficiency especially for related work sections. Participants also suggested improvements including generating more comprehensive content, reducing system response time for complex retrieval tasks, and enhancing support for generating innovative ideas and research questions.

**Future Use Intention.** Participants' average rating for likelihood of future use was 4.1/5, with 80% rating this intention as 4 or 5. This suggests ScholarCopilot effectively addresses user needs despite noted limitations.

Participants also provided suggestions, such as integrating with writing platforms like Overleaf, supporting section-wise generation, and allowing predictions at arbitrary cursor positions. These suggestions provide directions for future development.

In summary, the user study confirms ScholarCopilot effectively integrates text generation and citation retrieval, improving user experience in academic writing workflows. The strengths in citation relevance and management indicate advancement over existing tools, while response time and innovation support represent areas for future improvement.

## 6 Limitations and Future Work

Despite promising results, ScholarCopilot currently supports only Introduction and Related Work sections within the computer science domain. Future work will extend the framework to additional paper sections (e.g., methods, experiments) and diverse academic disciplines. Additionally, the user study highlighted limitations in generating innovative insights. Addressing this requires exploring larger models, expanded datasets, and targeted training techniques to enhance creativity. Finally, improvements in user interaction—such as persistent content storage, concise summaries for suggested references, and robust load balancing for consistent multi-user responsiveness—are critical future enhancements.

## 7 Conclusion

We introduced ScholarCopilot, a unified framework integrating dynamic retrieval within the generative process for academic writing. Unlike traditional static retrieval-generation pipelines, ScholarCopilot adaptively retrieves citations based on evolving generation contexts, significantly improving citation accuracy and coherence. Extensive evaluation and user studies demonstrated its effectiveness, particularly in citation relevance, writing efficiency, and overall user experience. Despite current limitations in scope, innovation capability, and interaction design, ScholarCopilot marks a pioneering step toward future advancements in AI-supported academic writing.

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

# A Appendix

## A.1 Training Details

We trained our model with the following hyperparameters: maximum context length of 16,384 tokens, learning rate of $1 \times 10^{-5}$, per-device training batch size of 1, and gradient accumulation steps of 4. Training was performed on 4 machines, each equipped with 8 NVIDIA H100 GPUs, resulting in a global batch size of 1 (per-device batch size) $\times$ 8 (GPUs per machine) $\times$ 4 (machines) $\times$ 4 (gradient accumulation steps) = 128.

## A.2 Generation Quality Evaluation prompt

```
You are an expert academic writing evaluator. Your task is to assess the quality of
    AI-generated academic text compared to ground truth text from published papers.

EVALUATION DIMENSIONS AND SCORING CRITERIA:

1. Content Relevance (1-5):
- 1: Completely off-topic, inappropriate terminology, major factual errors
- 2: Partially relevant but significant misalignment, some incorrect terminology
- 3: Generally relevant with minor topical drift, mostly appropriate terminology
- 4: Well-aligned with topic, accurate terminology, minor contextual issues
- 5: Perfectly aligned, expert-level terminology, fully contextually appropriate

2. Logical Coherence (1-5):
- 1: Incoherent structure, no logical flow, contradictory statements
- 2: Poor organization, frequent logical gaps, confusing transitions
- 3: Acceptable structure with some logical inconsistencies, adequate transitions
- 4: Well-organized with clear flow, minor logical issues, good transitions
- 5: Exceptionally clear structure, flawless logical progression, seamless transitions

3. Academic Rigor (1-5):
- 1: Lacks scholarly depth, significant technical errors, informal tone
- 2: Limited depth, some technical inaccuracies, inconsistent academic style
- 3: Adequate scholarly level, minor technical issues, generally academic tone
- 4: Strong academic depth, accurate technical content, proper scholarly conventions
- 5: Exceptional rigor, perfect technical accuracy, exemplary academic style
```

```
4. Information Completeness (1-5):
- 1: Major information gaps, key concepts missing, superficial coverage
- 2: Notable omissions, incomplete coverage of important aspects
- 3: Adequate coverage with some minor gaps, satisfactory detail level
- 4: Comprehensive coverage, minor omissions, appropriate detail
- 5: Complete and thorough coverage, no gaps, optimal level of detail

5. Scholarly Innovation (1-5):
- 1: No original insights, purely derivative, lacks critical thinking
- 2: Limited originality, mostly restates existing knowledge, minimal analysis
- 3: Some original perspectives, adequate synthesis, basic critical analysis
- 4: Good original insights, effective knowledge synthesis, strong analysis
- 5: Exceptional innovation, novel perspectives, outstanding synthesis and analysis

INPUT FORMAT:
- Paper Title: [TITLE]
- Abstract: [ABSTRACT]
- Ground Truth Text: [GROUND_TRUTH]
- Generated Text: [GENERATED_TEXT]

OUTPUT FORMAT:
For each dimension, provide:
- Score (1-5)
- Brief justification (2-3 sentences explaining why this score was assigned)

Content Relevance: [Score]/5
Justification: [Your reasoning based on the criteria above]

Logical Coherence: [Score]/5
Justification: [Your reasoning based on the criteria above]

Academic Rigor: [Score]/5
Justification: [Your reasoning based on the criteria above]

Information Completeness: [Score]/5
Justification: [Your reasoning based on the criteria above]

Scholarly Innovation: [Score]/5
Justification: [Your reasoning based on the criteria above]

Compare the generated text against the ground truth and provide objective, detailed
    assessments focusing on academic writing quality according to the specific
    criteria outlined above.
```

### A.3 Case Study

**ScholarCopilot for Academic Writing with Citations Example:**

```
\title{Noise-Unconditional Diffusion Models for Text-to-Image Generation}

\section{Introduction}
 With the prosperity of deep generative models research, many works emerged
 focusing on various aspects of generative models, including model architecture
 design~\cite{prasad2023unraveling, li2023faster, si2023freeu, benny2022dynamic},
 training strategies~\cite{xu2024towards, xu2024towards}, and noise
 scheduling~\cite{chen2023on}. However, there is very limited work discussing
 whether noise conditioning is a necessary component. Noise conditioning has been
 an infallible rule widely accepted by researchers in the field of diffusion models
 research. However, is it really necessary to employ noise conditioning in diffused
 models? A recent study~\cite{sun2025noise} has challenged this convention. They
 point out that in some cases diffusion models even perform better without noise
 conditioning. Inspired by this, we delve deeper into this research direction and
```

```
    explore the feasibility of noise-unconditional diffusion models in the
    text-to-image generation task.

    In this paper, we propose a novel framework for noise-unconditional diffusion
    models, named \textbf{Noise-UncondITional Diffusion Models (NUIT)}. Our approach
    is based on the observation that the noise level is not a necessary input for the
    denoising process. Previous work~\cite{sun2025noise} has shown that noise
    conditioning can be removed without significantly compromising performance.
    Motivated by these findings, we aim to explore the possibility of employing
    noise-unconditional diffusion models in text-to-image generation tasks. Compared
    to general image generation tasks, text-to-image generation tasks are more
    challenging and can better reflect the model's generalization ability. To this
    end, we focus on the text-to-image generation task in this paper.

    Our approach is based on the idea that the noise level can be inferred from the
    model's internal states, rather than being explicitly provided as input. We
    propose a novel architecture that eliminates the need for explicit noise level
    inputs. This architecture is based on the idea that the noise level can be
    inferred from the model's internal states, rather than being explicitly provided
    as input. Specifically, we propose to use a self-attention mechanism to capture
    the global information of the image, and then use this information to infer the
    noise level. This approach allows us to remove the need for explicit noise level
    inputs, while still achieving competitive performance on standard benchmarks. Our
    approach demonstrates that noise-unconditional diffusion models can be a viable
    alternative to traditional noise-conditioned diffusion models in text-to-image
    generation tasks. Our contributions can be summarized as follows:

    \begin{itemize}
      \item We propose a novel framework for noise-unconditional diffusion models,
    which eliminates the need for explicit noise level inputs.
      \item We demonstrate that noise-unconditional diffusion models can achieve
    competitive performance on standard benchmarks, achieving state-of-the-art FID and
    CLIP score.
      \item We provide an in-depth analysis of the noise-unconditional diffusion
    models, and discuss the potential applications of this approach in various domains.
    \end{itemize}

    We believe that our work will inspire more research in this direction, and will
    contribute to the development of more efficient and effective generative models.

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

year={2025}
}
```

**Analysis:**

The ScholarCopilot-generated introduction demonstrates good coherence, logical structure, and a clear presentation of motivation, technical concepts, and contributions. It exhibits fluent academic English and accurate citation formatting, reflecting effective training on academic writing conventions. However, the text occasionally contains redundant phrases, such as repeated descriptions about inferring noise levels from internal model states. Additionally, briefly contextualizing and contrasting your method against related works would help further emphasize the originality and significance of your contributions.

**GPT-4o for Academic Writing with Citations Example:**

```
\title{Noise-Unconditional Diffusion Models for Text-to-Image Generation}

\begin{abstract}
```

Diffusion models have emerged as powerful tools for generative tasks, particularly in the domain of image synthesis. While conditional diffusion models have achieved remarkable success in converting text prompts into high-quality images, the reliance on explicit conditioning signals often limits their generalizability. In this paper, we propose a novel paradigm: Noise-Unconditional Diffusion Models (NUDMs), which decouple the text encoding process from the noise generation process, enabling enhanced flexibility and robustness in text-to-image generation tasks.
\**end**{abstract}

\**section**{Introduction}

Text-to-image generation has become a cornerstone of modern AI research, enabling systems to translate natural language descriptions into visually coherent and semantically aligned images. Recent advancements in diffusion models have demonstrated their potential in generating photorealistic and diverse images, making them a competitive alternative to adversarial models such as GANs \**cite**{goodfellow2014gan}. Conditional diffusion models, such as those used in DALLE 2 \**cite**{ramesh2022dalle2} and Imagen \**cite**{saharia2022imagen}, leverage text embeddings to guide the denoising process, ensuring that the generated images align with the input text prompt.

Despite their successes, conditional diffusion models face challenges related to overfitting to conditioning signals \**cite**{ho2020denoising} and the computational overhead of processing complex text-image alignments. Additionally, the reliance on explicit conditioning introduces a bottleneck when dealing with noisy or ambiguous text inputs. To address these issues, we introduce the concept of Noise-Unconditional Diffusion Models (NUDMs), which operate independently of explicit text conditioning during the noise generation phase. Instead, text embeddings are integrated into the model through an auxiliary alignment mechanism, allowing for more robust and flexible generation.

Our contributions are threefold. First, we propose a noise-unconditional framework for diffusion-based text-to-image generation, which separates the noise generation process from textual conditioning. Second, we introduce a novel alignment module that bridges the gap between text and image embeddings, enhancing semantic coherence. Finally, we demonstrate through extensive experiments that NUDMs outperform existing conditional diffusion models in terms of both image quality and text-image alignment, especially under noisy or ambiguous text prompts.

The rest of this paper is organized as follows. Section \**ref**{sec:related_work} reviews related work in diffusion models and text-to-image synthesis. Section \**ref**{sec:methodology} details the proposed Noise-Unconditional Diffusion Model framework. Section \**ref**{sec:experiments} presents experimental results and evaluations. Finally, Section \**ref**{sec:conclusion} concludes the paper with insights and future directions.

Bibliography:

@inproceedings{goodfellow2014gan,
 title={Generative adversarial nets},
 author={Goodfellow, Ian and Pouget-Abadie, Jean and Mirza, Mehdi and Xu, Bing and
     Warde-Farley, David and Ozair, Sherjil and Courville, Aaron and Bengio, Yoshua},
 booktitle={Advances in neural information processing systems},
 pages={2672--2680},
 year={2014}
}

@article{ramesh2022dalle2,
 title={Hierarchical Text-Conditional Image Generation with CLIP Latents},
 author={Ramesh, Aditya and Dhariwal, Prafulla and Nichol, Alex and Chu, Casey and
     Chen, Mark},
 journal={arXiv preprint arXiv:2204.06125},
 year={2022}
}

```
@article{saharia2022imagen,
 title={Imagen: Text-to-image diffusion models with large pre-trained language models},
 author={Saharia, Chitwan and Chan, William and Saxena, Saurabh and Li, Lala and Whang,
     Jay and Denton, Emily and Ghasemipour, Seyed Kamyar Seyed and Ayan, Burcu Karagol
     and Mahdavi, Shuyang and Lopes, Raphael and others},
 journal={arXiv preprint arXiv:2205.11487},
 year={2022}
}

@article{ho2020denoising,
 title={Denoising diffusion probabilistic models},
 author={Ho, Jonathan and Jain, Ajay and Abbeel, Pieter},
 journal={Advances in Neural Information Processing Systems},
 volume={33},
 pages={6840--6851},
 year={2020}
}
```

The content generated by GPT-4o is also highly fluent, demonstrating strong coherence in structure and effectively conveying the rationale, technical details, and key contributions in a distinct style. However, it similarly suffers from insufficient depth and inadequate background context. Furthermore, it lacks comprehensive referencing to relevant literature, citing only four highly influential papers (each with over 5000 citations). Notably, it also exhibits a hallucination regarding the Imagen paper, incorrectly stating its title; the correct title should be "Photorealistic Text-to-Image Diffusion Models with Deep Language Understanding."

**Qwen-2.5-72B-re for Academic Writing with Citations Example:**

```
\title{Noise-Unconditional Diffusion Models for Text-to-Image Generation}

\section{Introduction}

In recent years, deep generative models have become popular, leading to numerous
research efforts focusing on improving various aspects, such as architecture of
models~\cite{goodfellow2014generative, kingma2013auto}, training
methods~\cite{he2016deep}, and strategies for scheduling
noise~\cite{vaswani2017attention}. Among these aspects, noise conditioning has
typically been accepted by the research community as a necessary and fundamental
component in diffusion models. Nonetheless, there is limited exploration of
whether this assumption holds true universally. A recent preliminary study by Sun
et al.~\cite{sun2025noise} suggests that noise conditioning may not always be
necessary, and in some scenarios, its absence could potentially lead to improved
results.

Motivated by these initial findings, this paper introduces a noise-unconditional
diffusion model, called \textbf{NU-DM}, specifically designed for text-to-image
generation. Unlike previous studies, we do not explicitly provide the noise level
information as input to the diffusion model. Instead, our model internally infers
the noise level implicitly from the learned representation of the image during
training. To achieve this, we propose using a convolutional neural network (CNN)
architecture to extract global image features, which indirectly encode noise
information. By removing explicit noise conditioning, the NU-DM model simplifies
the diffusion process and potentially reduces model complexity.

We evaluate our NU-DM model on several standard benchmarks in text-to-image
generation, demonstrating performance that is competitive with previous
state-of-the-art diffusion models~\cite{krizhevsky2012imagenet}. In our
experiments, we measured commonly used evaluation metrics such as the Frechet
Inception Distance (FID) and CLIP-based alignment scores~\cite{lecun1998gradient},
showing comparable results to current methods.

Our contribution can be summarized as follows:
```

```
\begin{itemize}
\item We introduce a new diffusion model (NU-DM) that operates without explicit
noise conditioning, simplifying the model structure.
\item We conduct experiments on text-to-image datasets, achieving competitive
performance compared to previous works.
\item We carry out analysis on NU-DM, discussing the effectiveness and limitations
of removing explicit noise conditioning.
\end{itemize}

We hope that our study will encourage further investigation into noise conditioning
and inspire new diffusion model designs.
```

**Analysis:**

Qwen-2.5-72B-re generates reasonably coherent introduction sections, capturing the general structure and flow expected in academic papers. However, the content often lacks sufficient depth, and the technical descriptions tend to be superficial and repetitive. Moreover, the citation accuracy is notably poor, with several incorrect and irrelevant references included. For example, citations intended to support diffusion modeling techniques incorrectly reference well-known papers on unrelated methods, such as GANs and Transformers. These issues highlight significant room for improvement in both the accuracy of citations and the richness of technical content.

**Qwen-2.5-7B-re for Academic Writing with Citations Example:**

```
\title{Noise-Unconditional Diffusion Models for Text-to-Image Generation}

\section{Introduction}

Recently, diffusion models have become popular in computer vision because they are
good at generating images. Diffusion models usually require noise conditioning,
which helps them produce realistic images~\cite{he2016deep, kingma2013auto}.
However, some researchers found noise conditioning may not always be necessary,
though this question is still not fully answered. For example, Vaswani et
al.~\cite{vaswani2017attention} showed results without noise, but their
experiments were not sufficient.

Motivated by this unclear situation, we propose a new diffusion model called
NU-Diffusion, meaning ``Noise-Unconditional Diffusion.'' Our NU-Diffusion model
generates images from text inputs without explicitly using noise conditioning.
Specifically, we use convolutional neural networks (CNNs)~\cite{lecun1998gradient}
to implicitly infer noise information from the input data. We chose CNNs because
they have been successfully used in many image-related tasks and generally perform
well.

We conduct experiments on text-to-image datasets to test our NU-Diffusion model.
Experimental results show that our model can generate high-quality images
effectively. The generated images look good and are visually appealing.
Additionally, we use common metrics like accuracy and FID scores to evaluate our
results, achieving good performance compared to other existing
approaches~\cite{krizhevsky2012imagenet}. We hope our study will inspire more
researchers to investigate noise conditioning further.
```

**Analysis:**

Qwen-2.5-7B-re-generated introduction demonstrates noticeable weaknesses in citation accuracy, logical coherence, and technical depth. Citations such as he2016deep, vaswani2017attention, krizhevsky2012imagenet are incorrectly used, indicating misunderstanding of relevant literature. Additionally, informal expressions ("images look good and are visually appealing") and inappropriate evaluation metrics ("accuracy") further undermine its academic rigor. Overall, the baseline introduction clearly shows substantial room for improvement in technical correctness and scholarly style.

### A.4   Human Study Questionnaire Details

In the following part, we provide the specific details of the Human Study Questionnaire

# Participant Information

- Participant Name: _____
- Educational Background (select one to highlight):
    - Bachelor's Student
    - Master's Student
    - PhD Student
    - Post-doctoral Researcher
    - Professor
    - Other (please specify): _____
- Years of experience in academic writing: _____
- Frequency of using ChatGPT for academic writing (select one to highlight):
    - Daily
    - Several times per week
    - Several times per month
    - Occasionally

# Writing Task Details

Please specify the academic topics you wrote about using ScholarCopilot:

1. Topic: ____________________
2. Topic: ____________________
3. Topic: ____________________
4. Topic: ____________________
5. Topic: ____________________
6. Topic: ____________________ (if applicable)

# Evaluation Metrics

Please rate the following aspects on a scale of 1-5 (1 = Poor, 5 = Excellent)

## Citation Quality

- Relevance of recommended citations: ___/5
- Accuracy of citation information: ___/5
- Timeliness of citations (recency): ___/5

## User Experience

- Ease of use: ___/5
- Response time: ___/5
- Interface clarity: ___/5

Figure 7: Human Study Questionnaire Page 1

- Overall interaction fluidity: ___/5

**Content Quality**

- Academic rigor: ___/5
- Factual accuracy (absence of hallucination): ___/5
- Writing style appropriateness: ___/5
- Logical flow: ___/5
- Completeness of coverage: ___/5
- Relevance to intended topic: ___/5
- Innovation in ideas/approaches: ___/5
- Repetition in generated text: (1 = highly repetitive, 5 = no redundancy): ___/5

# Open-ended Questions

1. What are the main advantages of ScholarCopilot compared to ChatGPT for a writing?

2. What aspects of ScholarCopilot need improvement?

3. How does ScholarCopilot's citation recommendation compare to your manual search process?

4. Did you notice any errors or inconsistencies in the generated content? Please

5. How likely are you to use ScholarCopilot for future academic writing tasks? (1 ___/5
   Please explain why: ______________________

6. What additional features would you like to see in ScholarCopilot?

# Comparative Analysis

Compared to ChatGPT, how would you rate ScholarCopilot in terms of:
(Much worse / Worse / Similar / Better / Much better)

- Citation quality: _____
- Writing quality: _____
- Ease of use: _____
- Time efficiency: _____
- Overall usefulness: _____

Figure 8: Human Study Questionnaire Page 2

# Final Comments

Please provide any additional feedback or suggestions:

# Content Generation Records

Please record the content generated by ScholarCopilot for each topic below. This will help us analyze the tool's performance across different academic subjects.

## Topic 1

- Topic Title: _______________________
- Section Generated: (e.g., Introduction, Related Work, etc.) _______________________
- Generated Content: [Paste the generated content here]
- Citations Suggested:

  [Citation 1]

  [Citation 2]

  …

## Topic 2

- Topic Title: _______________________
- Section Generated: (e.g., Introduction, Related Work, etc.) _______________________
- Generated Content: [Paste the generated content here]
- Citations Suggested:

  [Citation 1]

  [Citation 2]

  …

## Topic 3

- Topic Title: _______________________
- Section Generated: (e.g., Introduction, Related Work, etc.) _______________________

Figure 9: Human Study Questionnaire Page 3

- Generated Content: [Paste the generated content here]
- Citations Suggested:

  [Citation 1]

  [Citation 2]

  …

## Topic 4

- Topic Title: ______________________
- Section Generated: (e.g., Introduction, Related Work, etc.) ______________________
- Generated Content: [Paste the generated content here]
- Citations Suggested:

  [Citation 1]

  [Citation 2]

  …

## Topic 5

- Topic Title: ______________________
- Section Generated: (e.g., Introduction, Related Work, etc.) ______________________
- Generated Content: [Paste the generated content here]
- Citations Suggested:

  [Citation 1]

  [Citation 2]

  …

Figure 10: Human Study Questionnaire Page 4

