# OpenReview forum: "ScholarCopilot: Training Large Language Models for Academic Writing with Accurate Citations"
_colmweb.org/COLM/2025/Conference — COLM 2025_

### Official Review · Reviewer_T1dK · 2025-05-02

**Rating:** 6
**Confidence:** 3
**Ethics Flag:** 1

**Summary:**

This paper introduces ScholarCopilot for generating academic writing with large language. First, it contains they collect a large dataset from arXiv, where each paper is matched with its respective citation papers. The training process is consisted of a joint optimization process that leverages both a next token prediction loss and a contrastive loss. The evaluation is a sample from the original dataset where the model is asked to generate academic writing and the trained model is demonstrated to be much stronger than untrained baselines (Qwen-2.5-7/72B). The retrieval results are also better than the existing retrievers. Finally, the authors conduct user study to compare ScholarCopilot and ChatGPT and found notable improvements. While the paper studies an important task and introduces interesting training settings, it still lacks fair comparisons and validation of its evaluation setting.

**Questions To Authors:**

- Are there plans to release the dataset and model?
- What base model do you use for training?
- In the user study, does ChatGPT have access to the internet/search?

**Reasons To Accept:**

- The paper studies an important and realistic application of how to generating reliable academic writing with correct citations.
- The joint optimization technique is interesting and extend previous methods (e.g., GritLM) to a natural application.
- The results appear strong but still missing comprehensive baselines (explained below).
- The human study is useful in understanding how this framework may be used and received in real applications.

**Reasons To Reject:**

- The paper is missing important details and it can be difficult to understand the training and evaluation settings. Specifically, what are the inputs and outputs to the model during both training and evaluations? Does the model only need to generate the introduction paragraph (this is my impression from reading the examples in the appendix, apologies if I missed this in the text)?
- The evaluation setting is not rigorous—the paper leverages GPT-4o to evaluate the output on 5 axes, but lacks extensive analysis on how well GPT-4o performs as a judge in this setting. This is concerning as academic writing can be very long in length and require knowledge of recent papers that may be beyond the knowledge cutoff of the model. Using experts to validate the GPT-4o judge results would be much more convincing.
- The baselines are not fair for comparison—while the ScholarCopilot is trained on the collected dataset (which makes the evaluation in-domain) all other baselines are out-of-domain. It would be a more fair comparison to train both retrievers and the generator LMs (i.e., for the results presented in Table 1 and Figure 5) on the training set to understand how much benefit joint optimization brings.
- Due to the lack of rigor validating the evaluation setting, it may be convincing to use an existing benchmark for evaluation, such as ScholarQABench (Asai et al., 2024) or LitSeach (Ajith et al., EMNLP 2024) for evaluation.

Minor:
- missing discussion with previous citation works: Enabling Large Language Models to Generate Text with Citations (Gao et al., EMNLP 2023). also LitSearch (Ajith et al., 2024. EMNLP 2024) and CiteME (Press et al., 2024)

---

> ### Author Response · Authors · 2025-06-01
> **Response to Reviewer T1dK**
>
> ### 1. Clarification of Training and Evaluation Settings
> We appreciate your comments on the clarity of our training and evaluation pipeline. In fact, most of these details are already present in the manuscript. Specifically:
> - **Model Inputs/Outputs:** The inputs and outputs for both training and evaluation are described in several places. Figure 2 illustrates that the inputs include the paper title and abstract. Section 3.1 (line 128) specifies that our training data comprises structured fields: titles, abstracts, introductions, and related work sections.
> - **Evaluation Content:** Section 5.1 (line 248) states that both automatic and human evaluations are conducted on the introduction and related work sections.
> - **Scope Limitation:** Furthermore, in Section 6 (Limitations), we explicitly state that our current system supports only the generation of introduction and related work sections. We will further highlight and clarify these points in the revision to ensure there is no ambiguity.
>
> ### 2. Evaluation Rigor and Use of GPT-4o
> - While GPT-4o may have limitations regarding academic writing, we provide it with both the model output and the ground truth text for comparison (see Section 4.2, line 188). This setup allows GPT-4o to directly compare outputs and reduces risks related to hallucinations or outdated knowledge.
> - Additionally, as described in Section 5, we conducted a comprehensive human evaluation using domain experts, which helps to address potential limitations of automated LLM-based evaluation. We will further clarify this dual evaluation procedure in the revised manuscript to make our methodology more transparent.
>
> ### 3. Baseline Fairness
>
> Given our focus on real-world academic writing applications, we believe it is also reasonable to compare ScholarCopilot directly with existing solutions. Prior works such as GritLM and OneGen have demonstrated the advantages of unified generation and retrieval approaches, and our main goal is to showcase these benefits in the academic writing context. Nevertheless, we greatly appreciate your advice and will strengthen our experimental rigor accordingly.
>
> ### 4. Evaluation Benchmarks
>
> We appreciate the suggestion to use external benchmarks. As mentioned in the main text, ScholarCopilot currently operates in an autocomplete setting, comparable to a base model. We believe that further instruction tuning is needed to fully leverage its capabilities on benchmarks like ScholarQABench and LitSearch. We will consider this in future work and explicitly mention it as a direction for follow-up research in our revision.
>
> ### 5. Additional Details
>
> - We will add discussion of related works, including "Enabling Large Language Models to Generate Text with Citations" (Gao et al., EMNLP 2023), LitSearch (Ajith et al., 2024), and CiteME (Press et al., 2024).
> - Both the dataset and the model will absolutely be released to benefit the community.
> - The base model used is Qwen-2.5-7B.
> - In the user study, ChatGPT had access to internet search.

---

> > ### Author Response · Authors · 2025-06-05
> > **Follow-up on Our Response**
> >
> > Hi Reviewer T1dK,
> >
> > We would greatly appreciate any additional feedback or thoughts you may have on our responses. Please feel free to share any further questions or comments.

---

> > ### Comment · Reviewer_T1dK · 2025-06-05
> >
> > Thanks for the clarification on the training and evaluation settings as well as the additional details. To better understand how well GPT-4o serves as the judge in this case, it would be helpful to see its agreement with the human study conducted in Sec 5.
> >
> > In terms of baselines, the paper would be stronger if more existing systems were evaluated, which would enable further analysis. Specifically, the OpenScholar (Asai et al., 2024) retriever and model are highly relevant to this work, and evaluating them would give more insights into how the ScholarCopilot approach compares. The claim that OpenScholar "can lead to representational misalignment for implicit query intent and increased inference overhead" would be better supported by empirical results.

---

> ### Author Response · Authors · 2025-06-05
>
> Thank you very much for your valuable suggestions and feedback.
>
> As we noted in line 248 of the paper, the human evaluation study was conducted by inviting experts to use ScholarCopilot to write the introduction and related work sections for at least five self-selected topics. In contrast, the GPT-4o evaluation in Section 4.2 was conducted using 1,000 out-of-training-set arXiv papers as evaluation data. Since the data used in the human study and GPT-4o evaluation are different, it is not feasible to directly compare the agreement between GPT-4o and the human evaluations.
>
> Regarding the baseline suggestion, we appreciate your pointing out the relevance of OpenScholar (Asai et al., 2024). We are currently running experiments to evaluate OpenScholar’s retrieval accuracy on our 1,000 evaluation examples, both with and without a reranker. However, we would like to note that this comparison is inherently somewhat imbalanced: OpenScholar is designed for QA tasks and is trained on question-answering data, while our setting is autocomplete for scientific writing. Nevertheless, we will report OpenScholar’s retrieval scores later once the experiments are completed.

---

> > ### Author Response · Authors · 2025-06-10
> > **add openscholar eval results**
> >
> > As a follow-up, we have conducted additional experiments to include the performance of OpenScholar’s retriever, both with and without a reranker, on our 1,000 evaluation examples. Below are the results, which include retrieval accuracy at various ranks (R@k) for all models, including our ScholarCopilot system:
> >
> > | **Model**                   | **R@1** | **R@2** | **R@3** | **R@4** | **R@5** | **R@6** | **R@7** | **R@8** | **R@9** | **R@10** |
> > |-----------------------------|---------|---------|---------|---------|---------|---------|---------|---------|---------|----------|
> > | **BM25**                    | 9.8%    | 13.3%   | 15.3%   | 16.6%   | 17.6%   | 18.3%   | 19.1%   | 19.7%   | 20.2%   | 20.8%    |
> > | **E5-Mistral-7B**           | 15.0%   | 20.1%   | 22.5%   | 24.2%   | 25.5%   | 26.6%   | 27.6%   | 28.5%   | 29.3%   | 30.0%    |
> > | **OpenScholar_Retriever**   | 23.2%   | 27.5%   | 30.5%   | 33.0%   | 35.2%   | 37.1%   | 38.8%   | 40.4%   | 42.9%   | 45.3%    |
> > | **OpenScholar_Retriever + Reranker** | 24.6%   | 29.0%   | 32.3%   | 35.0%   | 37.4%   | 39.6%   | 41.6%   | 43.5%   | 46.0%   | 48.6%    |
> > | **ScholarCopilot**          | 40.1%   | 48.7%   | 53.3%   | 56.2%   | 58.4%   | 60.2%   | 61.6%   | 62.8%   | 63.9%   | 64.8%    |
> >
> > These results demonstrate that OpenScholar performs reasonably well in our evaluation setting, especially when combined with a reranker. However, as noted previously, OpenScholar is primarily designed for QA tasks and trained on question-answering data, whereas our task focuses on autocomplete for scientific writing.
> >
> > We hope this additional analysis addresses your concerns. Please feel free to discuss with us.

---

> > > ### Comment · Reviewer_T1dK · 2025-06-10
> > >
> > > Thanks for the additional experiments; it's interesting to see that the OpenScholar Retriever and Reranker underperform ScholarCopilot. My major concern is addressed and I have updated my score accordingly.

---

> > > > ### Author Response · Authors · 2025-06-10
> > > >
> > > > Thank you very much for your valuable suggestions and feedback.

---

### Official Review · Reviewer_UtQh · 2025-05-05

**Rating:** 3
**Confidence:** 4
**Ethics Flag:** 1

**Summary:**

The paper presents ScholarCopilot, an iterative RAG framework for scientific writing that dynamically retrieves reference papers when a special [RET] token is generated. The authors construct a large dataset of computer science papers and their citations to test the proposed framework. The experimental results show that ScholarCopilot outperforms the baseline models in an automated evaluation using GPT-4o, but it is not completely clear from the text how exactly the baselines work (see questions below). The authors also perform a human evaluation comparing ScholarCopilot to GPT-4o that includes user experience criteria.

**Questions To Authors:**

In Section 3.2, I am confused as to how the special retrieval tokens work. Once [RET] is generated, how is its initial embedding determined? Then, how is its embedding updated? Is there a special subnetwork dedicated to [RET] tokens, or are they being updated via the same self attention as the rest of the generation history? There is no mention of the former in the text, but the latter seems to me as though the contrastive loss and the generation loss could be in conflict.

What is the base model implementing ScholarCopilot? It is not mentioned in Sections 3, 4, or A1. I assume it is Qwen-2.5-7B, but this should be mentioned explicitly to ensure the comparison against the baselines is fair and reasonable.

Please explain more clearly what the experimental settings mean. For "Groundtruth Citations, is the model generating the non-citation text, with the ground truth citations inserted at the correct positions? For "Retrieved Citations," how do the baseline approaches decide when to retrieve a reference paper? Are they also using the [RET] token? Are they following a traditional RAG pipeline approach where the reference papers are retrieved before the generation step?

What exactly is the retrieved reference content? Is it the whole reference paper, or just the title/abstract/some other sections?

Why do the human evaluators not use the same evaluation criteria as the automated (GPT-4o) evaluation? I would have liked to see whether the GPT-4o evaluation correlated with human judgments, especially since the human evaluation compared ScholarCopilot against GPT-4o and found that GPT-4o performed worse --- how then can we trust the evaluation performed by GPT-4o?

**Reasons To Accept:**

The idea behind the approach is reasonable and well-motivated. The dataset could be useful to other researchers. However, this version of the manuscript is incomplete (see reasons to reject below). I have no problem with the proposed approach, and I can believe that it outperforms the baselines, but there is enough missing information that I could not reproduce this work, nor can I be sure that the evaluation is fair. It seems the authors' thoughts are running ahead of their words, and they leave out many details that may seem obvious to them, but the reader has no way of finding out.

**Reasons To Reject:**

A significant amount of information is missing from the main text of the paper, making it 1) impossible to reproduce and 2) difficult to determine the fairness of the evaluation. For example, what is the input to ScholarCopilot? Only from reading the prompts in the appendices can a reader learn that the input is the title and abstract of the paper whose Introduction/Related Work section is being generated. There are several other important details that are not found anywhere; see questions below. I do not believe this paper is publishable in its current form.

---

> ### Author Response · Authors · 2025-05-28
> **Response to Reviewer UtQh**
>
> Thank you for your review, though we respectfully disagree with both your assessment and rating. We believe many of your concerns stem from overlooking information that is clearly presented in our paper.
>
> ## Regarding [RET] Token Mechanism
>
> The mechanism of [RET] tokens is explicitly detailed in Section 3.2 and Figure 4. To clarify: we use a unified framework for generation and retrieval with Qwen-2.5-7B as our base model, jointly optimizing next token prediction loss and contrastive loss.
>
> Your main question appears to be about how the [RET] embedding is determined. As shown in Figure 4, we use the hidden state corresponding to the generated [RET] token as its embedding. In other words, we leverage the hidden states during generation as the token's representation. This approach has been established as effective in prior works such as GritLM and OneGen.
>
> ## Input to ScholarCopilot
>
> You claim information about model inputs is only available in the appendix, but this is incorrect. Figure 2 clearly shows that the model's input consists of the title and abstract. This is a central element of our main paper, not hidden in supplementary materials.
>
> ## Retrieved Reference Content
>
> Again, the answer to your question about retrieved reference content is plainly visible in Figure 2, where "gpt-4 reference" explicitly indicates that reference content consists of title + abstract. This information is presented prominently in the main text.
>
> ## Evaluation Methodology
>
> Regarding your concern about trusting GPT-4o for evaluation when it performed worse than ScholarCopilot: Please note that as we explained in Section 4.2 (line 188), GPT-4 as a judge evaluates based on the model output content and the article's ground truth text. GPT-4 simply scores according to ground truth and evaluation criteria. We understand your concern and will add more analysis on this point in our revised version.
>
> As for why human evaluators used different criteria than GPT-4o: our user study aimed to cover additional aspects like Open-ended Feedback, while the content quality questions in our survey largely covered the same areas as GPT-4o's evaluation. For your suggestion, we manually verified the quality of GPT-4 scoring on 50 samples and found the scoring quality to be high. We will add this information to the revised version.
>
>
> ## Experimental Settings
>
> For "Groundtruth Citations," we prompted baseline models to generate introductions and related work sections using provided relevant references along with article titles and abstracts. The prompt instructed models to insert accurate citations at appropriate positions - a strong baseline for traditional RAG methods since we provided ground truth references.
>
> For "Retrieved Citations," we prompted baseline models to generate placeholders where citations were needed, triggering external retrievers. Unlike "Groundtruth Citations," this approach doesn't follow traditional RAG methods but uses a generate-and-retrieve approach similar to ScholarCopilot.
>
> ## Base Model and Additional Details
>
> We confirm that we use Qwen-2.5-7B as our base model and will emphasize this and address other points you raised in our revised version.
>
> We appreciate your suggestions for clarification and will enhance the clarity of these details in our revision.

---

> > ### Author Response · Authors · 2025-06-05
> > **Looking Forward to Your Feedback**
> >
> > Hi Reviewer UtQh,
> >
> > We’d love to hear your thoughts or feedback on our responses. Please feel free to share any additional comments or questions you might have!

---

> > > ### Author Response · Authors · 2025-06-05
> > > **Follow-up on Our Response**
> > >
> > > To address your concerns about reproducibility, we have organized and anonymized the implementation code and uploaded it to an anonymous repository. You can access it at https://anonymous.4open.science/r/ScholarCopilotAnony-050F for more details that may help resolve any remaining reproducibility issues.

---

> > > > ### Author Response · Authors · 2025-06-07
> > > >
> > > > Does our response address your concerns? We’d love for you to join the discussion.

---

> > > > > ### Author Response · Authors · 2025-06-10
> > > > > **Awaiting Your Response to Our Rebuttal**
> > > > >
> > > > > We’ve carefully addressed your concerns in our rebuttal and would appreciate your follow-up. Since your review significantly influences the evaluation, we believe it is important that the discussion continues to ensure fairness and accountability.

---

### Official Review · Reviewer_caCT · 2025-05-12

**Rating:** 8
**Confidence:** 5
**Ethics Flag:** 1

**Summary:**

ScholarCopilot adopts a dynamic retrieve and generate pipeline to write academic texts with appropriate citations. It's a retrieval-augmented generation to mitigate hallucinations, but not static, as retrieval token generation is also within the responsibility of the developed system. The work releases a dataset of 500k computer science papers from arXiv, where all the contexts (with or without citations) are collected and all the citations are matched with papers in the academic databases.
ScholarCopilot jointly optimizes the next token prediction with citation retrieval to predict where in the text to cite and to whom. The authors also conducted a human evaluation to test the quality of academic writing against LLMs with retrieve-then-generate pipelines.

The results show that ScholarCopilot outperforms the baseline retrieval models in citation retrieval performance.
The ScholarCopilot has two variants: With and without reference content. The one with reference content generates more specific details when describing citations.
Human evaluation results confirm the system's usefulness in writing contexts with appropriate citations.

ScholarCopilot is less effective in proposing novel ideas and can only cite the publications in its reference base.

**Reasons To Accept:**

The work releases a dataset of 500k computer science papers from arXiv, where all the contexts (with or without citations) are collected and all the citations are matched with papers in the academic databases.
ScholarCopilot jointly optimizes the next token prediction with citation retrieval to predict where in the text to cite and to whom.
The results show that ScholarCopilot outperforms the baseline retrieval models in citation retrieval performance.
Human evaluation results confirm the system's usefulness in writing contexts with appropriate citations.

**Reasons To Reject:**

ScholarCopilot is less effective in proposing novel ideas and can only cite the publications in its reference base.

---

> ### Author Response · Authors · 2025-05-28
> **Response to Reviewer caCT**
>
> We sincerely thank the reviewer for the positive and encouraging assessment of our work on ScholarCopilot. We appreciate your recognition of our efforts in unifying generation and retrieval within academic text generation, as well as your acknowledgment of the value of the large, high-quality dataset we have curated. We believe this dataset will indeed be a valuable resource for future research in this area.
>
> Regarding the limitation that ScholarCopilot can only cite publications within its reference base, we would like to clarify that our system is designed to support efficient and regular updates to the corpus. New references can be incrementally encoded and added to the retrieval base, allowing the system to continually expand its citation capabilities with minimal effort. We believe this flexibility addresses the concern and ensures the system remains current and broadly useful.
>
> Thank you again for your thoughtful review and strong recommendation.

---

> > ### Comment · Reviewer_caCT · 2025-06-04
> > **Acknowledgement**
> >
> > Thank you for your response.

---

### Official Review · Reviewer_mo4f · 2025-05-13

**Rating:** 6
**Confidence:** 5
**Ethics Flag:** 1

**Summary:**

This paper presents ScholarCopilot, a fine-tuned language model for generating academic articles. The model is trained end-to-end to retrieve relevant papers in context, by generating a special [RET] token; the representation of the special token is used to perform nearest neighbor search over embeddings of articles from a database; then the retrieved paper's information is provided in-context to enable a RAG-style generation.

The authors first collected a large dataset of academic articles by parsing and cleaning arxiv, with bib entries linked to other arxiv or semantic scholar articles. The authors then performed end-to-end training with both next-token-prediction loss on the article content and a standard contrastive learning loss for the retrieval part.

The other contribution is the evaluation: the authors proposed to use some of the inline citation mentions for retrieval evaluation and to use GPT-4o for a generation quality vibe checking. The proposed system outperformed existing retrieval-methods or standard RAG systems (with much bigger LLMs). The human evaluation also supports a similar conclusion.

**Questions To Authors:**

Please see "reasons to reject".

Another question: is it correct that [RET] is always placed at where inline citation mentions are? In some cases, the citation is placed at the end of the sentence and hence will be too late if retrieval happens there (e.g., "GPT-3 shows large language model can do in-context learning [cite GPT-3]"). In fact, I think the majority of the citations look like that? Also, how does the training data handle the case where there are multiple citations, like "Pre-trained language models are powerful [cite BERT; cite GPT-3]".

**Reasons To Accept:**

(1) The proposed method (both training and the system) is very interesting (especially the inline retrieval part). The evaluation also shows that this system performs significantly better than ad-hoc RAG systems or standalone embeddings.

(2) The application that this paper is targeting could be very impactful and hasn't been studied much. To the COLM community, this direction is worth exploring.

(3) One of the biggest contributions of this paper is the cleaned data with citations linked to arxiv/semantic scholar entries. Subsequent works could significantly benefit such data.

(4) The paper itself is well written; for such a new application, the authors provided a complete solution (data, method, evaluation, human evaluation).

**Reasons To Reject:**

My biggest complain is about the evaluation:

(1) For the generation part: the definitions of the GPT-4o evaluation metrics are very vague (even for the prompt in A.2, there is only a few key words like "content relevance" without explicitly explaining what that means). There is also an important part missing---citation quality (which was included in the human evaluation)---which not only includes the retrieval accuracy (which was benchmarked separately) but also how well the cited paper supports the claim. For a work that spends a lot of effort on the inline citation training, this seems to be a huge oversight for the evaluation part. There have already been many studies on how to automatically evaluate inline citations, such as [ALCE](https://arxiv.org/abs/2305.14627).

(2) For the retrieval part: it makes sense to use inline citation mentions for testing since that's what the application of paper writing requires. However, existing benchmarks that utilize inline citation mentions (for example, [Gu et al., 2022](https://arxiv.org/pdf/2112.01206)) are often very noisy and newer works in citation recommendation use more complicated pipelines to clean up queries ([LitSearch](https://arxiv.org/abs/2407.18940), [CiteMe](https://arxiv.org/abs/2407.12861)). It's unclear how the authors pick those testing instances (maybe I missed it). Also, is it ensured that those examples are not relevant to the training data in any sense (for example, they are not even used as retrieval positive examples).

In general, there is a lack of discussion on several relevant papers on citation, generation evaluation, and academic paper retrieval/recommendation. I don't think it's necessary for this work to use existing established methods as it is a novel application, but there should at least be some discussions on how they are different.

---

> ### Author Response · Authors · 2025-05-28
> **Response to Reviewer mo4f**
>
> Thank you for your thoughtful review and valuable feedback. We appreciate your positive assessment of our work on ScholarCopilot, particularly highlighting the interesting method, potential impact, and the contribution of our cleaned dataset with linked citations.
>
> Regarding your concerns about evaluation:
>
> ## Citation Quality and Retrieval Evaluation
>
> We conducted an ablation study on Reference Content Integration (Section 4.4.2) which demonstrates that retrieved reference content effectively improves generation quality, particularly in "Coherence" and "Academic" aspects. This partially addresses your concern about citation quality, though we acknowledge we could provide more extensive evaluation.
>
> You raise an important point about works like ALCE, LitSearch, and CiteMe. While our model is fine-tuned specifically for academic writing and may not be directly comparable on these benchmarks, we agree that discussing these differences would strengthen our paper. We will add comprehensive comparisons and discussions of these related works in the revised version.
>
> Regarding your question about testing instance selection: we carefully ensured test examples were not used as retrieval positive examples during training to avoid data contamination. We will clarify our selection methodology in the revision.
>
> ## [RET] Token Placement and Multiple Citations
>
> To address your question about [RET] token placement: ScholarCopilot can indeed generate the [RET] token at any position in the sentence, not just where inline citations would traditionally appear. The model learns to place these tokens based on its training data patterns.
>
> You raise a valid concern about late retrieval (e.g., at sentence end) potentially limiting the usefulness for enhancing generation. However, as supported by our ablation study in Section 4.4.2 (Table 2), even without using retrieved content, the model's base generation quality remains high (average score of 15.53, close to Qwen-2.5-72B's 15.81), indicating strong inherent knowledge.
>
> Regarding multiple citations: our model can generate multiple [RET] tokens where appropriate, not just a single retrieval. We included examples demonstrating this capability in Appendix A.3's case study. Our training data contains numerous instances of multiple citations, which the model learned to emulate.
>
> ## Evaluation Metrics Clarity
>
> We acknowledge your feedback that the GPT-4o evaluation metrics definitions are insufficiently detailed. In our revision, we will provide more comprehensive explanations of these metrics and include additional case studies to illustrate the evaluation process.
>
> Thank you again for your constructive feedback. We believe addressing these points will significantly strengthen our paper.

---

> > ### Comment · Reviewer_mo4f · 2025-06-02
> > **Thank you for the response**
> >
> > Thank you for the response! The 4.4.2 content indeed verifies the effectiveness of the retrieved content. The additional discussions and clarifications will strengthen the paper significantly!

---

### Decision · Program_Chairs · 2025-07-08

**Decision:**

Accept

**Comment:**

This paper presents a full LLM pipeline for generating academic research articles. The contributions include an end-to-end training approach for both retrieval and generation, a large-scale training dataset constructed by parsing and cleaning arXiv, and an automatic evaluation based on GPT-4o. I find the topic highly interesting to the COLM community and potentially impactful, and the contributions are solid. Therefore, I recommend the acceptance of the paper.

Several concerns were raised by the reviewers, and I strongly recommend that the authors address them in the final revision (mostly related to writing and clarity):
* The definitions of the GPT-4o evaluation metrics are insufficiently detailed, and it remains unclear how reliable GPT-4o is as a judge for this challenging task.
* The inputs and outputs to the models and evaluators can be defined clearer. The authors should provide a rigorous textual description rather than relying solely on figures.
* The comparison to OpenScholar is interesting and should be included in the final version.
* Please discuss other relevant work, such as ALCE, CiteMe, and LitSearch.

[Automatically added comment] At least one review was discounted during the decision process due to quality]